# Genetic Dissection of Phosphorus Use Efficiency in a Maize Association Population under Two P Levels in the Field

**DOI:** 10.3390/ijms22179311

**Published:** 2021-08-27

**Authors:** Dongdong Li, Haoying Wang, Meng Wang, Guoliang Li, Zhe Chen, Willmar L. Leiser, Thea Mi Weiß, Xiaohuan Lu, Ming Wang, Shaojiang Chen, Fanjun Chen, Lixing Yuan, Tobias Würschum, Wenxin Liu

**Affiliations:** 1Key Laboratory of Crop Heterosis and Utilization, the Ministry of Education, Key Laboratory of Crop Genetic Improvement, Beijing Municipality, National Maize Improvement Center, College of Agronomy and Biotechnology, China Agricultural University, Beijing 100193, China; dongdongli@cau.edu.cn (D.L.); wanghaoying98@163.com (H.W.); wm1212131@163.com (M.W.); guoliangli@cau.edu.cn (G.L.); xiaohuanlu111@163.com (X.L.); wangming7273@163.com (M.W.); shaoj@cau.edu.cn (S.C.); 2Key Laboratory of Plant-Soil Interaction, the Ministry of Education, Center for Resources, Environment and Food Security, College of Resources and Environmental Sciences, China Agricultural University, Beijing 100193, China; chenz9418@163.com (Z.C.); caucfj@cau.edu.cn (F.C.); yuanlixing@cau.edu.cn (L.Y.); 3State Plant Breeding Institute, University of Hohenheim, 70593 Stuttgart, Germany; willmar_leiser@uni-hohenheim.de (W.L.L.); theami.weiss@uni-hohenheim.de (T.M.W.); 4Institute of Plant Breeding, Seed Science and Population Genetics, University of Hohenheim, 70593 Stuttgart, Germany; tobias.wuerschum@uni-hohenheim.de; 5Institute of Crop Science, Chinese Academy of Agricultural Sciences, Beijing 100081, China

**Keywords:** maize, P stress, phosphorus use efficiency, genome-wide association study, gene ontology analysis, phylogenetic characterization

## Abstract

Phosphorus (P) deficiency is an important challenge the world faces while having to increase crop yields. It is therefore necessary to select maize (*Zea may* L.) genotypes with high phosphorus use efficiency (PUE). Here, we extensively analyzed the biomass, grain yield, and PUE-related traits of 359 maize inbred lines grown under both low-P and normal-P conditions. A significant decrease in grain yield per plant and biomass, an increase in PUE under low-P condition, as well as significant correlations between the two treatments were observed. In a genome-wide association study, 49, 53, and 48 candidate genes were identified for eleven traits under low-P, normal-P conditions, and in low-P tolerance index (phenotype under low-P divided by phenotype under normal-P condition) datasets, respectively. Several gene ontology pathways were enriched for the genes identified under low-P condition. In addition, seven key genes related to phosphate transporter or stress response were molecularly characterized. Further analyses uncovered the favorable haplotype for several core genes, which is less prevalent in modern lines but often enriched in a specific subpopulation. Collectively, our research provides progress in the genetic dissection and molecular characterization of PUE in maize.

## 1. Introduction

Maize is a multi-purpose crop, being an important staple food, feed, and industrial raw material. It is very important to ensure stable maize production. Phosphorus (P) is a major element necessary for plant growth and development. P deficiency is an important abiotic stress that limits crop yield [1,2]. Around the world, cropland in many regions is in a state of P deficiency, especially in developing countries [3]. Generally, farmers solve the problem of soil P deficiency by applying chemical fertilizers. However, excessive application and P deposition into rivers has led to the low use efficiency of P fertilizers, which causes many ecological and environmental problems [4,5]. At the same time, phosphate rock, a non-renewable resource, is expected to be depleted in the next 100–400 years [6]. Therefore, it is critical to identify genotypes that have high P use efficiency (PUE, defined as the yield produced per unit of P available in the soil) and thus obtain a higher yield under P deficient conditions, by applying molecular breeding methods based on genetic analysis of PUE [7].

Under P stress, plants show a series of P starvation responses to ensure normal growth. For example, plants change root structure, membrane structure, release more organic acids and enzymes like phosphatases and phytases, and regulate the expression of P starvation response genes [2,8], involving the transcriptome, proteome, metabolome, and other levels. This eventually leads to changes in the physiological and morphological characteristics of plants, such as the accumulation of anthocyanins [9,10], root characteristics [11,12], plant height [13,14], biomass, and yield [13,15]. Therefore, the absorption and utilization of P in plants is regulated by a complex genetic network. These changes at the phenotypic and molecular level are the basis for analyzing the genetic mechanism of PUE. At the same time, plant height, biomass, and yield are suitable for screening potential low-P tolerant genotypes, and for genetic analyses.

In *Arabidopsis*, rice, and other crops, great progress has been made in understanding the mechanism of P regulation. For example, phosphate transporters are responsible for phosphate uptake and allocation [16]; *ARF7* and *ARF19* positively regulate the P starvation response [17]; *PHR1* regulates lipid remodeling during P starvation [18]; *ZAT6* synchronizes P homeostasis and root development [19]; *PSTOL1* is an enhancer of early root growth [20]; *BHLH32* has a negative effect on anthocyanin accumulation, root hair formation, and cellular P concentration [21,22]; the miR399–*PHO2* pathway [23] regulates the distribution of P in plants. In maize, some microRNA, such as Zma-miR3 [24] and miR399 [25], were identified in seedling roots under low-P condition; *PILNCR1*–miR399 interaction and the miR399–*PHO2* regulatory pathway have different modes of action in maize inbred lines with different PUE [26]; transcription factor *NIGT1.2* increases the absorption of P but decreases the absorption of ${\text{NO}}_{3}^{-}$ during P starvation, similar to the regulatory pathway in *Arabidopsis* [27]; transcription factor *ZmPHR1* can improve PUE under P-deficient conditions [28]. However, most genes were identified or confirmed through reverse genetics. Hence, there is an urgent need to identify the genes underlying natural variation through forward genetics.

Many quantitative trait loci (QTL) related to PUE have been detected in a wide range of genetic populations [29] using root-related traits [30,31,32], biomass-related traits [33], and yield-related traits [34,35]. Among them, the research and utilization of the locus *Pup1*, named *phosphorus-starvation tolerance 1 (PSTOL1)*, was the most successful, and its overexpression was shown to increase the biomass and yield of rice under low-P condition [20]. The homologous regions of rice *PSTOL1* in sorghum increase biomass and have an important role in the root system under low-P condition [36], and also play a similar part in maize [33]. In maize, Zhang et al. [34] mapped a major QTL *qKN* controlling kernel number per row under different P conditions; Qiu et al. [37] identified QTL *AP9*, which is related to acid phosphatase activity under P deficiency; and Mendes et al. [38] explored the genetic architecture of PUE in tropical maize using recombinant inbred lines (RILs).

In recent years, genome-wide association study (GWAS) has become a powerful tool for analyzing the genetic basis of complex traits in maize [39]. Xu et al. [35] conducted a GWAS of 18 traits under two P levels combined with RNA-seq data of extreme genotypes under P stress and identified 259 candidate genes, mainly involved in four pathways: transcriptional regulation, reactive oxygen scavenging, hormone regulation, and remodeling of the cell wall. Luo et al. [40] conducted their GWAS on 22 root-related phenotypes under two P levels at the seedling stage and combined them with the metabolomic data of extreme inbred lines; through this, three potential candidate genes GRMZM2G039588, GRMZM5G841893, and GRMZM2G051806 were found, and several were confirmed in RILs. Similarly, Wang et al. [31] used 13 root-related phenotypes at different P levels at the seedling stage to perform GWAS combined with transcriptome data and identified the candidate gene GRMZM2G009544, which is closely associated with several root traits, namely total root length, root forks (a measure of root branching), and the total number of root tips. These previous studies have shown that GWAS has great potential to be used in excavating candidate genes for maize P-related traits. The above studies mainly focused on root architecture or some visible traits at the seedling stage; however, for the internal P utilization in shoot, kernel, and grain yield at maturity stage in the association panel, to our knowledge, there is no report in maize until now.

In our study, the GWAS was conducted for an association panel containing 359 inbred lines for PUE, biomass, and yield-related traits under normal-P and low-P treatments in the field. The main objectives were as follows: (1) to explore the variation and relationship of traits under normal-P and low-P environments, (2) to identify candidate genes related to P stress response and PUE, (3) to identify conserved protein motifs and cis-elements of major candidate genes, and (4) to identify favorable haplotypes of genes related to PUE. Our research showed abundant and stable genetic variation in traits under P deficiency. Dozens of genes were identified by GWAS. Furthermore, several key candidates were mined and haplotype frequencies were explored to provide a strategy for molecular breeding of P-efficient lines.

## 2. Results

### 2.1. Phenotypic Variation under Low-P and Normal-P Conditions

Traits of an association panel consisting of 359 inbred lines were recorded under low-P and normal-P treatments in the field with the available P concentration in the soil around 2.1 mg/kg and 4.5 mg/kg, respectively. Wide variations of eleven traits were observed under the two P treatments (Table 1). Significant differences under low-P and normal-P conditions were detected for all traits. (Figure 1 and Appendix A). P utilization efficiency (PUtE) had an increase, especially shoot P utilization efficiency (ShPUtE) which increased by 55.0%; however, the remaining traits showed a significant decrease when comparing low-P to normal-P conditions. The various impact of P on plant growth was well shown in two traits. Yield per plant (YPP) was reduced by 68.3%, whereas seed P concentration (SePCc) showed only a slight reduction in low-P versus normal-P condition. The correlations of all traits were positively significant between the two P conditions. The highest correlation was observed for shoot dry weight per plant (SDWPP) (*r* = 0.69, *p* < 0.01), and the lowest for seed P utilization efficiency (SePUtE) (*r* = 0.36, *p* < 0.01). For YPP, this correlation was 0.67 (*p* < 0.01).

Under the low-P condition, the genetic variances of yield-related and biomass-related traits were generally increased (Table 1) compared to the normal-P condition regarding the genetic coefficient of variation (GCV). The variance of genotype-by-P treatment interaction ($\sigma_{G\times T}^{2}$) was only significant (*p* < 0.01) for SePCc, shoot P concentration (ShPCc), SePUtE, and ShPUtE. For all traits, the genetic variance component $\sigma_{G}^{2}$ was larger than the $\sigma_{G\times T}^{2}$. The repeatability was slightly higher under normal-P compared to the low-P condition for most traits, ranging from 0.34 for SePUtE under low-P to 0.72 for ShPCc under normal-P (Table 1). For YPP, the repeatability was 0.67 and 0.57 under low-P and normal-P conditions, respectively. The genetic variance across both P conditions was significant (*p* < 0.01) for all traits, and broad-sense heritability ranged between 0.47 for SePUtE to 0.77 for SDWPP. The heritability of YPP was high, reaching 0.69.

The correlations among eleven traits in low-P, normal-P, and the derived low-P tolerance index (LPTI) (estimated as phenotype under low-P divided by phenotype under normal-P) datasets were used to perform correlation analysis (Figure 2 and Appendix A). In the low-P dataset (Figure 2A), ShPUtE was significantly correlated with biomass (*r* = 0.2 for SDWPP and *r* = 0.24 for all dry weight per plant (ADWPP)) and with YPP (*r* = 0.34), which was also significantly correlated with SePUtE (*r* = 0.3) and seed P content per plant (SePCPP) (*r* = 0.3). For YPP under the low-P condition, SePUtE showed a significantly positive correlation (*r* = 0.31), but ShPCc and SePCc both showed significantly negative correlations, −0.3 and −0.3, respectively. In the normal-P dataset, ShPUtE also significantly correlated with YPP, SDWPP, and ADWPP, with coefficients 0.38, 0.19, and 0.35, respectively (Figure 2B). In the LPTI datasets, ShPUtE showed a slightly positive correlation (*r* = 0.2) with YPP and SDWPP (*r* = 0.11), but was not significantly positively correlated with ADWPP (Appendix A). In addition, we also observed a significant correlation between ShPUtE and SePUtE under the low-P condition (Figure 2A), but this could not be seen in the normal-P (Figure 2B) and LPTI dataset (Appendix A). A high correlation between different traits meant they had similar change patterns under each condition. In addition, significant correlations between ShPUtE with yield and biomass highlighted its important role.

### 2.2. Traits Distribution of Different Subpopulations under Low-P Condition

Based on a previous study [41], our 359 maize inbred lines were clustered into four subpopulations, namely non-stiff stalk (NSS), stiff stalk (SS), tropical and subtropical (TST), and mixed. To explore the phenotypic differences of these four groups under the low-P treatment, multiple comparisons were conducted. For SePUtE, there was not a significant difference among the four subpopulations. For ShPUtE and all P utilization efficiency (APUtE), the SS subpopulation showed significantly higher trait values than NSS, and slightly but not significantly higher values than TST and mixed subpopulations, which means that those lines within SS generally use the absorbed P in a more efficient way than other subpopulations under P deficient conditions (Figure 3). The TST subpopulation showed higher values for ADWPP (Appendix A) and SDWPP (Appendix A), which indicates that this group generally produced more biomass under low-P conditions, pointing to a higher P uptake efficiency. For the remaining traits, there was no significant difference (Appendix A). Therefore, these two subpopulations have the potential to breed lines with high P uptake efficiency and P utilization efficiency.

### 2.3. Genome-Wide Association Study to Identify P-Stress Responsive Genes

Genome-wide association study (GWAS) was performed on the traits with the low-P, normal-P, and LPTI datasets (Appendix A). We identified 92, 72, and 63 significant SNPs for eleven traits in the three datasets, respectively, and 49, 53, and 48 candidate genes were identified in total (Table 2 and Appendix A). Gene annotations were obtained from MaizeGDB (https://www.maizegdb.org/, accessed on 9 June 2021). Besides, their homologous genes in *Arabidopsis* were obtained through BLASTP, and annotations were downloaded from TAIR (https://www.arabidopsis.org/, accessed on 9 June 2021). Seven key genes related to phosphate transporter proteins, abiotic stress response, root architecture, or members of the BHLH and F-BOX gene family are listed in Table 2 and the rest are listed in Appendix A.

A significant SNP, namely chr5.S_31881708 (−log_10_(*P*) = 5.0), was found for ShPUtE under the low-P condition (Figure 4A,B). This SNP is at the exon of the gene GRMZM2G326707 (Figure 4C), and there was a significant difference (*p* < 0.01) between the two genotypes (Figure 4D). GRMZM2G326707 (*ZmPHT1;1*) encodes the phosphate transporter protein1 (PHT1) in maize [42]. The *PHT1* gene family was widely studied in many crops and observed to play an important role in the phosphorus starvation response and PUE regulation [25,43,44,45]. Furthermore, four significant SNPs were found for ShPUtE under low-P condition, which were all located in the exon of the GRMZM5G848945 gene region (Appendix A). GRMZM5G848945 encodes protein F-BOX3, which is important in root development and stress responses [46,47]. Interestingly, there were two significant SNPs identified for LPTI of SePCc, and another gene, GRMZM2G155849, encoding the F-BOX3 protein was found (Appendix A). In addition, the significant SNP chr8.S_162559636 found for SePCPP under low-P condition was located in the gene GRMZM2G030762 (Appendix A), which encodes the transcription factor *bHLH55*, a member of BHLH family. In maize, transcription factor *bHLH55* can enhance plant salt stress through regulation of the biosynthesis of ascorbic acid, which is an antioxidant and enzyme vital to abiotic stress tolerance [48].

Besides, several genes were identified across different datasets. For example, common genes GRMZM2G171254 and GRMZM2G171277 were identified for several traits, particularly ShPUtE and ShPCc under the low-P condition, and APUtE, ShPCc, ShPUtE and YPP under the normal-P condition. In addition, GRMZM2G084296 was identified for SDWPP under low-P condition and SePUtE for LPTI.

### 2.4. Gene Ontology Analysis

The gene ontology (GO) enrichment analysis is an important method to understand gene functions. With a significant threshold of 4, a total of 397 genes were identified in the low-P dataset. These genes were found to be involved in two significant GO terms (*p* < 0.05) in the biological process, namely “response to abiotic stimulus” and “stomatal movement”, which showed that genes identified under the low-P condition are related to stress response (Figure 5A). Additionally, the other four significant terms in the cellular component, namely “organelle”, “intracellular organelle”, “membrane-bounded organelle”, and “intracellular membrane-bounded organelle”, are closely related to organelle (Figure 5B). The above results show that stress response and organelle functional genes play an important role in the plant P starvation response, which provided more information to confirm candidate genes.

### 2.5. Phylogenetic Characterization and Cis-Elements Prediction of PHT1 Gene Family

GRMZM2G326707 (*ZmPHT1;1*) encoding a phosphate transporter protein and a member of the *PHT1* gene family was identified in ShPUtE under the low-P condition. Phosphate transporters are responsible for phosphate uptake and allocation in the plant [16], and play an important role under low-P stress [49]. We collected 46 possible members of the *PHT1* gene family, 13 genes in maize (*Zea mays* L.), 13 genes in rice (*Oryza sativa* L.), 11 genes in sorghum (*Sorghum bicolor* L.), and 9 genes in *Arabidopsis* (*Arabidopsis thaliana* L.). A neighbor-joining tree was constructed using protein sequences (Figure 6). In total, 10 conservative motifs were identified. The PHT1 proteins shared high similarity and common motifs. Clear subfamilies could not be observed in spite of the species. In the dicot *Arabidopsis*, several genes were grouped into a subgroup, clearly showing the difference in differentiation between dicot and monocot species. In addition, the cis-elements analysis taking the DNA sequences upstream 1 kb of the transcriptional start site as a target identified some cis-elements related to hormones and stress. For example, ARE is involved in abscisic acid responsiveness, LTR is involved in low-temperature responsiveness, and TC-rich repeats are involved in defense and stress responsiveness.

### 2.6. Identification of Favorable Haplotype for Molecular Breeding among Different Subpopulations

In order to establish a guide for screening P-efficient materials and illustrate the frequency distribution of favorable haplotypes in different subpopulations, we selected five important P starvation-related genes. Among these genes, GRMZM2G326707 and GRMZM5G848945 were candidate genes found for ShPUtE under low-P condition by GWAS performed in our study, while the others were homologs of P starvation-related genes or root architecture-related genes in *Arabidopsis* found in previous studies; GRMZM2G381709 is a homolog of *PHO2* [50] related with Pi uptake and allocation remobilization, GRMZM2G088487 is a homolog of *ARP6* [51] as an epigenetic modulator of some P-starvation response genes, and GRMZM2G054050 is a homolog of *LPR1* [52] associated with root architecture [53]. Considering the important role of the ShPUtE part of PUE and significant correlations between yield and biomass (Figure 2), ShPUtE variations under the low-P condition of different haplotypes of five important genes were investigated (Figure 7A). Except for GRMZM2G381709, significant differences between favorable and the remaining haplotypes (*p* < 0.05) were observed. The proportions of favorable haplotypes varied between subpopulations and genes (Figure 7B). Generally, the proportions of favorable haplotypes were small, which indicated that there is still much space for improvement in modern inbred lines. For gene GRMZM2G054050, the SS subpopulation showed a large proportion of favorable haplotype, while the proportion of favorable haplotype in other subpopulations was relatively small. The distribution of the favorable haplotype of GRMZM2G381709 was also interesting, as it had a medium-to-high proportion in TST, mixed, and SS, but a small proportion in the NSS subpopulation (Figure 7A,B). Population differentiation resulted in the frequency of favorable genes among the different groups, and some germplasm resources harboring favorable haplotypes could be used to improve the other inbred lines.

## 3. Discussion

PUE is an important and complex trait, which can be defined by grain yield per available P in soil and hence is affected by many genetic factors. PUE consists of two molecular processes, Pi uptake from soil to root and shoot called P uptake efficiency (PUpE), and internal Pi utilization named P utilization efficiency (PUtE). PUE (yield/P_soil available_) = PUpE (P_t_/P_soil available_) × PUtE (yield/P_t_), where P_t_ is the total P in plant, and P_soil available_ is the available P in soil [15]. If using hydroponics, where precise P is applied, the PUE can be calculated accurately for each genotype. Different from that, in this study we recorded traits in the field, where the PUE cannot be calculated accurately due to the heterogeneity of soil and other uncontrollable factors. However, PUtE can be calculated precisely by using a chemical method. Therefore, we mainly focused on PUtE, a component of PUE. Dissecting the genetic architecture of PUE and mapping potential genomic loci associated with PUE is the first step for a molecular breeding program. As far as we know, our study is the first large-scale evaluation and genetic dissection of PUE-related traits in a diverse set of maize.

### 3.1. Phenotypic Variation for PUtE-Related, Yield and Biomass Traits under Two-P Levels

Low-P stress has a great impact on plant growth and development (Figure 1). To explore the effects of low-P on maize, we mainly focused on: (1) biomass and yield-related traits, including ADWPP, SDWPP, and YPP; and (2) PUE-related traits, including SePUtE, ShPUtE, and APUtE. Total biomass (ADWPP) and grain yield (YPP) decreased under low-P condition by 36% and 68%, respectively (Table 1). Such strong yield decreases show that there was a heavy P stress in our field. Thus, the genetic findings of this study may reflect actual PUE under severe P deficiencies and may offer guidance to further understand PUE under severely depleted soils. Similar strong P effects have been shown by several authors. Cai et al. [54] observed a significant decrease in maize grain yield under low phosphorus conditions, with the yield of RILs reduced by about 37% compared to the control. Chen et al. [55] found a significant decrease in maize shoot dry weight and shoot total P accumulation. Similar to Yao et al. [56], we also observed an increased PUtE under low-P versus normal-P condition. This showed that the available P was more efficiently used under low-P condition versus well fertilized conditions. However, there was also a significant genetic variation for PUtE under the normal-P condition, pointing to the potential of selecting material, which utilizes the supplied P much better and hence could increase PUtE under well fertilized conditions as well, without sacrificing much yield (Figure 1). The correlation analysis showed the positive correlation coefficients between ShPUtE and biomass, as well as yield-related traits, which made it possible to select lines with high yield and high PUE simultaneously. In practice, lines with high PUE under different P levels should be selected by plant breeders. We chose 20 lines with the highest APUtE under the low-P condition as P-efficient lines, namely “CIMBL91”, “CIMBL106”, “DAN599”, “GEMS56”, “CIMBL15”, “CML415”, “07KS4”, “CIMBL95”, “CIMBL38”, “GEMS14”, “GEMS24”, “GY220”, “WH413”, “975-12”, “GEMS23”, “CIMBL93”, “GEMS9”, “R08”, “YU374”, and “177”. Among these 20 lines, “CML415”, “CIMBL91”, and “07KS4” also stood in the top 20 for APUtE under the normal-P condition. Among these lines, there were six from NSS, three from SS, seven from TST, and four from the mixed subpopulation. In conclusion, PUE could be further improved in modern inbred lines. However, one needs to keep in mind that in order to translate it to actual PUE under farmers’ conditions in which hybrid varieties normally grow, further evaluations of test-cross performances and the inheritance patterns of the determined traits are necessary.

### 3.2. Key Candidate Genes for Low-P, Normal-P, and LPTI Datasets

With the rapid development of sequencing technology, GWAS is becoming increasingly important for dissecting the genetic architecture of complex traits. As an allogamous crop, maize has a comparably rapid linkage disequilibrium (LD) decay and abundant diversity, so the mapping resolution can reach down to the gene level underlying QTL with millions of SNPs [57]. In our study, several key genes were identified that were also reported in former studies. The F-BOX3 protein encoded by the GRMZM5G848945 gene is an auxin receptor. Its homologous gene in rice is *OsAFB2*. The downregulation of *OsAFB2* reduces the tolerance to salt stress and the sensitivity to auxin [58]. The homologous genes *AFB2* and *AFB3* in *Arabidopsis* are also closely related to plant abiotic stress [59]. GRMZM2G030762 encodes the transcription factor *bHLH55*. The bHLH family participates in multiple biological processes in plants [60], especially playing an important role in dealing with drought stress [61,62].

The homologous gene of the candidate gene GRMZM2G109967 identified under normal-P in *Arabidopsis* is AT2G39290, which encodes phosphatidylglycerolphosphate synthase 1, being essential for the biosynthesis of phosphatidylglycerol in chloroplasts [63,64]. There is no doubt that AT2G39290 is important in maintaining the normal function of chloroplasts, and GRMZM2G109967 may have a similar function in maize with the relationship of phospholipids biosynthesis. For this, it is easy to understand that GRMZM2G109967 is associated with ADWPP. The homologous gene of GRMZM2G418916 in *Arabidopsis* is *ROOT HAIR DEFECTIVE4*, which relates to the normal development of plant root hairs [65]. The root hairs of plants play an important role in the absorption and utilization of nutrients [66], and GRMZM2G418916 corresponds with SePCPP. Both genes are important for the normal growth and development of plants.

We also identified some key candidate genes in the LPTI dataset. Among them, the homologous gene of GRMZM2G076630 in *Arabidopsis* is *SLK2*, related to the process of embryogenesis [67] and the response to abiotic stress [68]. GRMZM2G104125 encodes calcium-dependent protein kinase 2 (CDPK2), while the homologous gene in *Arabidopsis* encodes CDPK19. CDPKs play an important role in plant growth and development, stress response, and signal transduction [69]. Interestingly, phospholipids can regulate the activity of CDPK [70], so the environment of different phosphorus levels may affect the effect of CDPK. LPTI showed the phosphorus stress tolerance of plants, so genes corresponding with stress response would be concerned. These key candidate genes were reported to be functional under other stress conditions, and we think they are also associated with P stress based on our results. Besides the above key genes, some genes in Appendix A might be valuable in the future when progress has been made in functional genomics in maize, rice, and *Arabidopsis*.

### 3.3. The Motif Compositions and Cis-Elements of the PHT1 Gene Family

GRMZM2G326707 (*ZmPHT1;1*) encodes a PHT1 protein, which was associated with ShPUtE under the low-P condition. Previous studies have shown that the function of PHT1 is to transport phosphate into the maize shoot [16]. *ZmPHT1;1* plays an important role in Pi uptake and redistribution in maize, and is induced during Pi starvation [49]. Besides, *AtPHT1;4* promoter stimulates reporter gene expression in the monocot root system under low-P condition [71]. Most *PHT1* genes are mainly expressed in roots and are upregulated under phosphorus starvation conditions [72]. Overexpression of Os*PHT1;4* in rice increased phosphorus accumulation in plant roots and shoots [73]. The above result shows that the *PHT1* family play an important role in the transport of Pi and the P starvation response in the plant. Therefore, it is necessary to explore the homologous genes and the conservative domain of the PHT1 protein family across the different crops.

With BLASTP using the protein sequence of GRMZM2G326707 (*ZmPHT1;1*) in maize as a query sequence, 13 genes were found in maize (*Zea mays* L.), 9 genes in *Arabidopsis* (*Arabidopsis thaliana* L.), 13 genes in rice (*Oryza sativa* L.), and 12 genes in sorghum (*Sorghum bicolor* L.). Based on a former study [74], a sorghum gene that did not contain the PHT1 specific signature (GGDYPLSATIxSE) was eliminated; in the end, 46 genes were left (Appendix A). Generally, ten conservative motifs were identified in most proteins, and the distribution of motifs was also similar. The phylogenetic tree revealed that many shared motifs existed in the PTH1 sequence and a highly conservative protein sequence, especially in the dicot crop *Arabidopsis*. Previous studies have also mentioned that the protein sequences of the *PHT1* genes were similar, and the expression patterns overlapped [75]. It also showed that the function of the *PHT1* family was probably realized by these motifs, which lays the foundation for searching for the functional site of GRMZM2G326707.

Cis-elements play an important role in gene regulation [76] and gene action. The promoter sequences of these genes identified some wound and abiotic stress-responsive cis-elements, such as ABRE, LTR, TC-rich repeats, and other hormone-responsive cis-elements. It illustrated that these genes were induced by abiotic stresses such as drought and hypoxia, and plant hormones such as abscisic acid and methyl jasmonate. Previous studies confirmed that some members in *Arabidopsis* were induced under the low-P condition [75], which pointed out the direction toward functional research on the members of the *PHT1* gene family.

### 3.4. Imbalanced Distribution of Favorable Haplotypes among Different Subpopulations

Maize originates from Mexico and has formed different groups under the domestication of humans, which shaped the wide phenotypic variation. During domestication, the frequency of favorable genetic fragments or genes changed due to genetic drift, mutation, and selection.

Many genes related to P uptake, utilization and translocation have been verified in many plants [49,77,78]. Therefore, it is very attractive to figure out the relationship between haplotype variation and PUE in the modern inbred lines. To explain this answer, in this study, two key genes identified in our GWAS and three published genes were taken as an example to define favorable haplotypes. One of five genes (GRMZM2G381709) was not significantly different between the favorable and the remaining haplotypes. Population structure may be one reason for this, which may explain why the gene was not identified in our GWAS. Some favorable genes were present at low frequency in all subpopulations but showed a high frequency in one subpopulation (Figure 7). This imbalanced distribution of favorable haplotypes illustrated that the proportion of some alleles could be further improved, and some alleles may be fixed in some subpopulations. Excellent lines with favorable haplotypes could be selected to improve other resources by introgression.

### 3.5. Breeding P-Efficient Maize Lines

For most traits, a moderate to high heritability can be observed (Table 1) under two P treatments, which provides potential to improve germplasm resources by the molecular breeding method. Through forward genetics, natural variation can be mined, and superior alleles could be aggregated, which can boost breeding efficiency. In rice [20], sorghum [36], and maize [33], protein kinase PSTOL1 confers higher plant biomass and yield under P stress. Considering several key candidates related to abiotic stress and calcium-dependent protein kinase 2 identified in our GWAS results (Table 2), we believe these genes are of importance in PUE breeding. Additionally, it should be noted that one specific subpopulation harboring a relatively high percentage of a superior allele for target genes can be chosen as a donor parent to improve efficiency (Figure 7). Furthermore, whole genomic selection as a powerful tool for molecular breeding has been widely used in animal and plant breeding. The prediction abilities were illustrated for all traits in three datasets. In the low-P dataset, the prediction ability ranged from 0.15 for APUtE to 0.54 for SDWPP. For grain yield, it can still yield a prediction ability of around 0.4. In the normal-P dataset, the prediction ability ranged from 0.10 for SePCc to 0.47 for SDWPP, while in the LPTI data the prediction ability ranged from −0.08 for ShPUtE to 0.23 for SDWPP (Appendix A). The result of the genomic selection study showed its feasibility and prospect in plant breeding, especially under P deficient conditions.

## 4. Materials and Methods

### 4.1. Plant Materials

A diverse GWAS population [41] was chosen and 359 lines were randomly selected for the study. It was further divided into 4 subgroups, including 28 stiff stalk lines, 113 non-stiff stalk, 111 tropical/subtropical, and 87 mixed lines [41].

### 4.2. Field Design

In May 2018, the panel was planted under the low-P and normal-P conditions in Shangzhuang Station of China Agricultural University, Beijing. P fertilizers had not been applied to the low-P field since 1985, but 45 kg/ha P_2_O_5_ was applied before sowing for the normal-P field every year. Additionally, 240 kg/ha N fertilizer was applied in both trials before planting [79]. Based on a former study [79], for the low-P trial, the N and K concentration were 0.63–0.83 mg/kg and 109.2–147.9 mg/kg, respectively; for the normal-P trial the N and K concentration were 0.69–0.77 mg/kg and 135.6–140.3 mg/kg, respectively. Before sowing, the Olsen P in the low-P and normal-P trial were measured using a NaHCO_3_ method [80] by taking nine samples uniformly from the 0–20 cm soil; the average value was 2.1 mg/kg and 4.5 mg/kg, respectively. The concentration of P was the main limiting factor. All other management measures remained the same. Each treatment was laid out as an augmented α-design, including three replicates. Each replicate included 16 blocks, and each block contained 25 plots with the check line ‘Ye478′. Each genotype was planted in a single row with a length of 1.2 m, a plant spacing of 0.2 m, and a row spacing of 0.5 m. Four plants in the middle of each plot were used for phenotypic measurements.

### 4.3. Acquisition of Traits

The traits in this study mainly include two aspects: (1) the yield-related and biomass-related traits: yield per plant (YPP), shoot dry weight per plant (SDWPP), all dry weight per plant (ADWPP); (2) PUE-related traits in seed and shoot, seed P concentration (SePCc), shoot P concentration (ShPCc), seed P content per plant (SePCPP), shoot P content per plant (ShPCPP), all P content per plant (APCPP), seed P utilization efficiency (SePUtE), shoot P utilization efficiency (ShPUtE), and all P utilization efficiency (APUtE).

Seed and shoot were harvested separately in the field. The harvested seed and shoot were dried in an oven at 65°C to a constant weight, and weighed to calculate the dry weight per plant and the yield per plant. Then, a high-speed pulverizer was used to pulverize the seed and shoot into powder, and about 0.2 g of seed and about 0.4 g of the shoot were weighed and digested with H_2_SO_4_ and H_2_O_2_ until the liquid was transparent. Then, the P concentration was measured following the spectrophotometric method.

P content is equal to the product of P concentration and dry weight, calculated in shoot and seed respectively; all P content per plant was calculated as APCPP = SePCPP + ShPCPP [38]. PUtE is equal to the inverse of P concentration [81], meaning the yield or dry matter mass produced by absorbing 1 mg of P, which represents the utilization efficiency of the phosphorus absorbed by the plant. The LPTI was calculated by the performance under low-P divided by the trait performance under normal-P, and was used as an index for GWAS. The descriptions of all traits used in this study are listed in Appendix A.

### 4.4. Phenotypic Data Analysis

First of all, the method of Studentized Residual Razor was used to remove outliers in the original data, with a threshold of 2.8 [82]. The best linear unbiased estimator (BLUE) value of each trait under low-P and normal-P conditions were calculated with the following formula:*y* = *μ* + *G* + *Rep* + *Block*(*Rep*) + *ε*,(1)
where *y* represented the phenotype observation value; *μ* was the overall mean; *G* was the genotypic effect; *Rep* was the effect of replication; *Block*(*Rep*) was the block effect nested within the replication; *ε* was the error, and *ε* was subject to follow a normal distribution in each replication; *G* was a fixed effect; and the rest were random effects.

For the analysis of genetic variance and interaction variance across the two P conditions, the model was:*y* = *μ* + *G* + *T* + *G*T* + *Rep*(*T*) + *Block*(*Rep*) + *ε*,(2)
where *y*, *μ*, *G*, *Block*(*Rep*) was the same as the above model; *T* was the effect of treatment with two P conditions; *G*T* was the interaction of the genotype-by-P treatment; and *Rep*(*T*) was the replication effect in each treatment, assuming that *ε* followed a normal distribution within each P treatment. Except for the *T* effect, the others were treated as random effects. The method of Cullis was used to calculate the repeatability under each P treatment and the heritability across treatments with the following formula [83]:(3)$$H^{2}=1-\frac{{\bar{\upsilon }}_{\mathrm{BLUP}}}{2\sigma_{G}^{2}},$$
where ${\bar{\upsilon }}_{\mathrm{BLUP}}$ was the mean variance of a difference of two BLUP, and $\sigma_{G}^{2}$ was the genetic variance estimated by REML in the R package ‘sommer’ (version 4.1.3) [84].

### 4.5. Genome-Wide Association Study

By integrating RNA-Seq data of 368 inbred lines and Illumina SNP50 Bead Chip genotype data of 513 inbred lines, 556,809 high-quality SNP data were obtained [85], publicly available at http://www.maizego.org/Resources.html (accessed on 9 June 2021). The reference genome in this study was B73 RefGen_v3. Based on the original 513 lines, the genotypes of 359 individuals were extracted and filtered according to the missing rate lower than 0.2 and the minor allele frequency greater than 0.05, and finally, 534,772 SNP markers remained. The Bayesian-information and Linkage-disequilibrium Iteratively Nested Keyway model, in which pseudo QTNs were used to control false positives and reduce false negatives [86], was used to implement a GWAS in GAPIT (version 3) [87]. Since redundant markers are in strong linkage disequilibrium, it was too strict to calculate the Bonferroni-corrected threshold using all markers. Therefore, we used the indep-pairwise module of PLINK (http://pngu.mgh.harvard.edu/purcell/plink/, version 1.9) [88] to calculate the independent marker numbers, with the parameters window size equal to 50, step size equal to 50, and *r^2^* greater than or equal to 0.2 [89]; finally, 87,096 independent markers were obtained. The suggestive threshold to control the type I error rate was global α = 0.10, thus the significant threshold was −log_10_(0.10/(87,096/10)) = 4.94 with chromosome-wide Bonferroni correction.

Candidate genes were identified when the significant SNPs were in genes or genes were within a 5 kb distance from the significant SNPs. Gene annotations were downloaded from maizeGDB (https://www.maizegdb.org/, accessed on 9 June 2021). For the key genes without annotations, BLASTP was conducted in Tbtools (version 1.075) [90] to get the best hit genes in *Arabidopsis*. One or two genes with the smallest e-value were taken as the homologous genes, and annotations were downloaded from TAIR (https://www.arabidopsis.org/, accessed on 9 June 2021).

### 4.6. Gene Ontology Analysis

To further understand the metabolic pathways of candidate genes, we conducted GO analysis for genes identified in GWAS in the low-P condition with a threshold of 4.0, which was slightly lower than the threshold of independent GWAS. The moderate threshold was used to balance the false positives and false negatives for the entries of GO analysis, which included another significance test to promise low false positives. This process was implemented using agriGO v2.0 (http://systemsbiology.cau.edu.cn/, version 2.0, accessed on 9 June 2021) [91].

### 4.7. Phylogenetic Characterization and Conserved Motif Analysis

To find out the conserved motifs of the PHT1 family among different plant species, the protein sequence of PHT1 in maize was used as a query sequence for BLAST. We identified *PHT1* homologous genes in four species, namely maize (*Zea mays* L.), *Arabidopsis* (*Arabidopsis thaliana* L.), rice (*Oryza sativa* L.), and sorghum (*Sorghum bicolor* L.). Corresponding gene annotations were obtained from the maizeGDB, TAIR, China Rice Data Center (http://www.ricedata.cn/gene/, accessed on 9 June 2021), and NCBI (https://www.ncbi.nlm.nih.gov/, accessed on 9 June 2021), respectively. One sorghum gene did not contain the PHT1 specific signature (GGDYPLSATIxSE) [74]. Then, protein sequences were downloaded from EnsemblPlants (http://plants.ensembl.org/, accessed on 9 June 2021). A phylogenetic tree was generated by mega7.0 [92] software, and motif analysis was completed with MEME by setting the maximum number of motifs to 10 (https://meme-suite.org/meme/, accessed on 9 June 2021). The DNA sequences 1 kb upstream of the transcriptional start site of the first transcript were extracted in TBtools and were submitted to PlantCARE (http://bioinformatics.psb.ugent.be/webtools/plantcare/html/, accessed on 9 June 2021) for prediction of cis-elements. Finally, the TBtools were used to repaint the results with default parameters.

### 4.8. Haplotype Identification

To further understand the differences between the haplotypes of target genes, and illustrate the percentage of favorable haplotypes among different subpopulations (SS, NSS, TST, and mixed), five genes, namely GRMZM2G326707, GRMZM5G848945, GRMZM2G381709, GRMZM2G088487, and GRMZM2G054050, closely related to P stress from both the results of GWAS in this study and previous studies [53], were taken as examples. Firstly, the SNPs located in the gene were chosen, then Tag SNPs were identified by setting *r*^2^ equal to 0.8 and the others as default in HaploView (version 4.2.) [93] Afterward, Tag SNPs were used to make up haplotypes among the population. To ensure the accuracy of statistics, we removed individuals and genotypes with missing phenotypes, and only major haplotypes (frequency > 0.05) were kept. Among these haplotypes, the one with the highest average effect was regarded as the favorable haplotype, and the others were classified as remaining haplotypes.

### 4.9. Genomic Selection

The genomic best linear unbiased prediction model [94] was implemented in R (version 4.0.3) package rrBLUP (version 4.6.1) [95] for genomic selection. The prediction ability was evaluated by the correlation between the actual values and predicted values. Five-fold cross-validation with 1000 repetitions was used to yield the final accuracy.

## Figures and Tables

**Figure 1 ijms-22-09311-f001:**
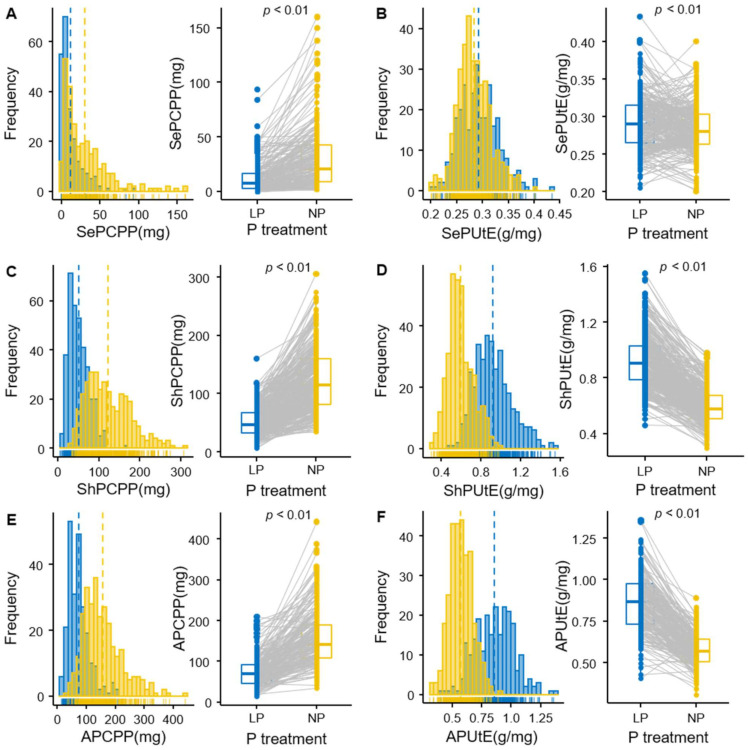
Distribution and interaction of traits under low-P and normal-P treatments. (**A**) SePCPP. (**B**) SePUtE. (**C**) ShPCPP. (**D**) ShPUtE. (**E**) APCPP. (**F**) APUtE. SePCPP: Seed P content per plant; SePUtE: Seed P utilization efficiency; ShPCPP: Shoot P content per plant; ShPUtE: Shoot P utilization efficiency; APCPP: P content per plant; APUtE: All P utilization efficiency. The significant difference was calculated by *t*-tests between the phenotypes under low-P and normal-P conditions for all traits.

**Figure 2 ijms-22-09311-f002:**
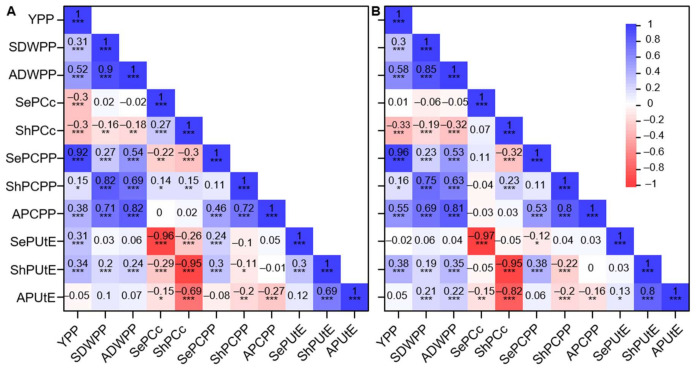
Correlations among eleven traits under low-P (**A**) and normal-P (**B**) conditions. YPP: Yield per plant; SDWPP: Shoot dry weight per plant; ADWPP: All dry weight per plant; SePCc: Seed P concentration; ShPCc: Shoot P concentration; SePCPP: Seed P content per plant; ShPCPP: Shoot P content per plant; APCPP: P content per plant; SePUtE: Seed P utilization efficiency; ShPUtE: Shoot P utilization efficiency; APUtE: All P utilization efficiency. *: *p* < 0.05; **: *p* < 0.01; ***: *p* < 0.001.

**Figure 3 ijms-22-09311-f003:**
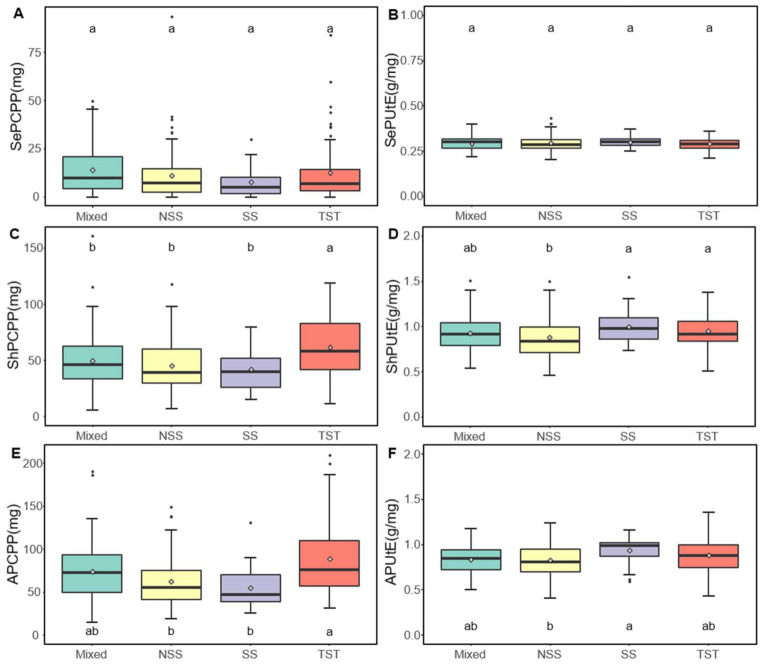
Boxplots for the performance of the four subgroups for six traits under low-P conditions. (**A**) SePCPP. (**B**) SePUtE. (**C**) ShPCPP. (**D**) ShPUtE. (**E**) APCPP. (**F**) APUtE. These four subpopulations are mixed (*n* = 101), non-stiff stalk (NSS, *n* = 118), stiff stalk (SS, *n* = 29), tropical/subtropical (TST, *n* = 111). Multiple comparisons were conducted by the LSD test method at a 0.05 significance level. Different letters represent significant differences. The rhombus in the boxplot is the mean value. SePCPP: Seed P content per plant; SePUtE: Seed P utilization efficiency; ShPCPP: Shoot P content per plant; ShPUtE: Shoot P utilization efficiency; APCPP: P content per plant; APUtE: All P utilization efficiency.

**Figure 4 ijms-22-09311-f004:**
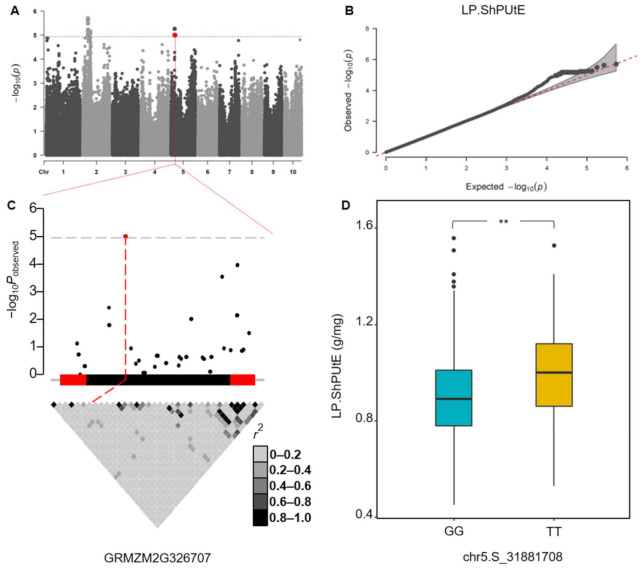
Manhattan (**A**) and quantile–quantile (**B**) plot for ShPUtE under low-P condition; gene structure of GRMZM2G326707 and pairwise linkage disequilibrium (LD) analysis (**C**); and the distribution of two genotypes for ShPUtE under the low-P condition in our population (*t*-test, *p* < 0.01) (**D**). The dotted line is the significance threshold of 4.94. ShPUtE: Shoot P utilization efficiency. **: Significant at 0.01 level.

**Figure 5 ijms-22-09311-f005:**
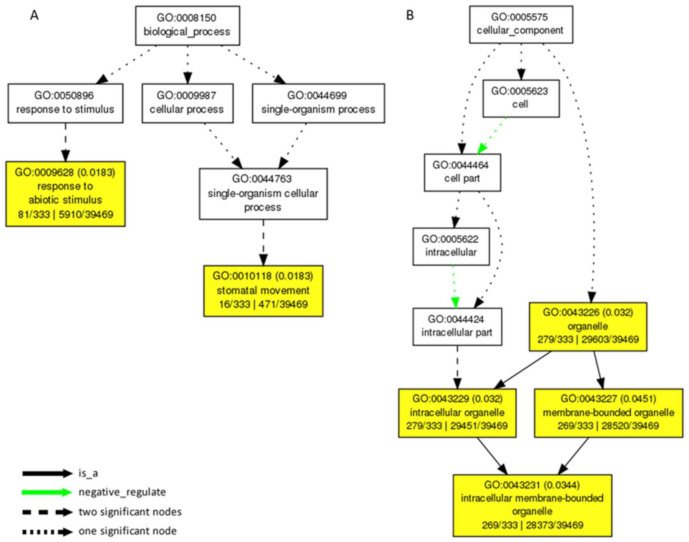
The significantly (*p* < 0.05) enriched GO terms for genes identified in GWAS under low-P condition with a significance threshold of 4. (**A**) Significant GO terms in the biological process. (**B**) Significant GO terms in the cellular component. The yellow rectangles represent significant GO terms (*p* < 0.05), and the green lines represent negative regulation.

**Figure 6 ijms-22-09311-f006:**
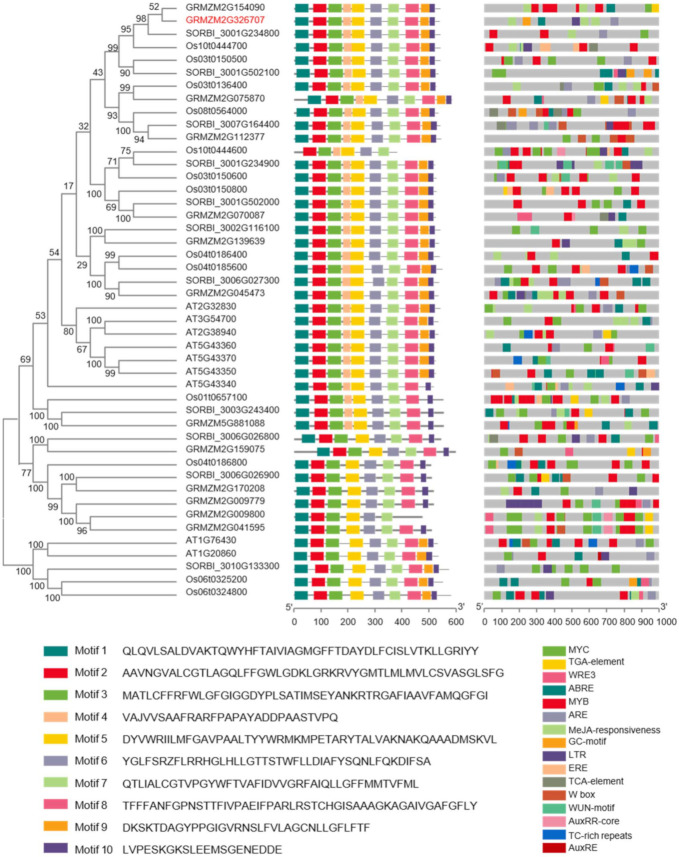
Polygenetic tree of *PHT1* gene family and distribution of conserved motifs and potential cis-elements. The gene GRMZM2G326707 (*ZmPHT1;1*) in red was identified in our GWAS result.

**Figure 7 ijms-22-09311-f007:**
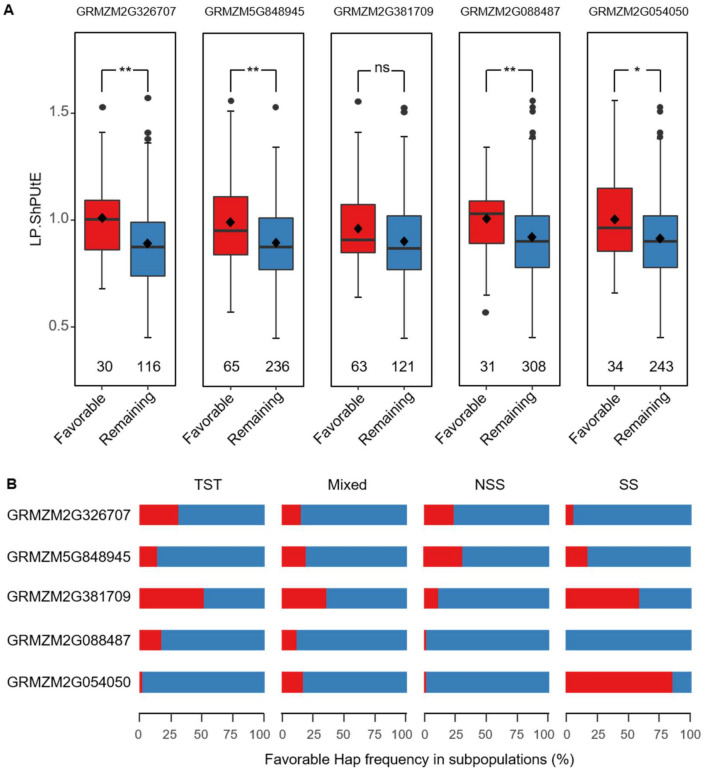
ShPUtE variation under low-P condition and frequencies of the favorable haplotypes of five important low-P responsive genes. (**A**) Box plots for ShPUtE under low-P condition of favorable and remaining haplotypes. The numbers in the box plot represent the individual number. The significance test was done by *t*-test. *: Significant at 0.05 level; **: Significant at 0.01 level; ns: No significant difference. (**B**) The proportion of two types of haplotypes in four subgroups. Red represents the favorable haplotype and blue represents the remaining haplotype. ShPUtE: Shoot P utilization efficiency.

**Table 1 ijms-22-09311-t001:** Summary statistics of all traits under the two P conditions.

Traits	P-Level	Mean	SD	*r*	Rd (%)	$\sigma_{G}^{2}$	*Rep* ^2^	$\sigma_{G-acr}^{2}$	$\sigma_{G\times T}^{2}$	GCV	*H* ^2^
YPP	LP	2.47	3.55	0.67 **	68.3	46.7 **	0.67	11.0 **	0.00	2.764	0.69
YPP	NP	7.80	7.91			68.1 **	0.57			1.058	
SDWPP	LP	46.7	21.5	0.69 **	34.0	501.1 **	0.64	338.9 **	9.38	0.479	0.77
SDWPP	NP	70.8	26.9			701.9 **	0.64			0.374	
ADWPP	LP	54.3	27.9	0.62 **	36.3	1022.2 **	0.68	523.4 **	14.27	0.589	0.72
ADWPP	NP	85.2	33.6			1280.7 **	0.64			0.420	
SePCc	LP	3.46	0.45	0.42 **	3.49	0.085 **	0.41	0.061 **	0.016 **	0.084	0.50
SePCc	NP	3.58	0.40			0.076 **	0.45			0.077	
ShPCc	LP	1.13	0.22	0.56 **	35.7	0.029 **	0.61	0.031 **	0.011 **	0.149	0.65
ShPCc	NP	1.76	0.37			0.10 **	0.72			0.182	
SePCPP	LP	12.0	14.0	0.66 **	60.0	395.9 **	0.53	153.1 **	0.80	1.658	0.53
SePCPP	NP	30.0	29.3			918.8 **	0.60			1.010	
ShPCPP	LP	51.5	24.5	0.55 **	58.2	463.9 **	0.50	353.4 **	0.00	0.419	0.63
ShPCPP	NP	123.2	52.8			1704.5 **	0.48			0.335	
APCPP	LP	73.2	37.0	0.52 **	53.3	1238.5 **	0.53	1001.9 **	127.9	0.481	0.54
APCPP	NP	156.7	67.9			4276.4 **	0.57			0.417	
SePUtE	LP	0.29	0.04	0.36 **	−3.31	0.00063 **	0.34	0.00038 **	0.00014 **	0.086	0.47
SePUtE	NP	0.28	0.03			0.00062 **	0.50			0.088	
ShPUtE	LP	0.92	0.19	0.56 **	−55.0	0.024 **	0.43	0.0099 **	0.0028 **	0.166	0.70
ShPUtE	NP	0.60	0.13			0.011 **	0.60			0.178	
APUtE	LP	0.86	0.17	0.46 **	−49.4	0.013	0.50	0.0057 **	0.00038	0.130	0.62
APUtE	NP	0.57	0.10			0.0065 *	0.65			0.135	

SD: Standard deviation; *r*: Correlation coefficient between low-P (LP) and normal-P treatment (NP); Rd (%): Relative reduction under low-P stress calculated by (mean (NP)-mean (LP))/mean (NP); *Rep*^2^: Repeatability in each treatment; $\sigma_{G-acr}^{2}$: Genetic variance across both P conditions; GCV: Genetic coefficient of variation calculated as sqrt($\sigma_{G}^{2}$)/mean; *H*^2^: Broad-sense heritability. *: Significant at 0.05 level, **: Significant at 0.01 level. YPP (g): Yield per plant; SDWPP (g): Shoot dry weight per plant; ADWPP (g): All dry weight per plant; SePCc (mg/g): Seed P concentration; ShPCc (mg/g): Shoot P concentration; SePCPP (mg): Seed P content per plant; ShPCPP (mg): Shoot P content per plant; APCPP (mg): All P content per plant; SePUtE (g/mg): Seed P utilization efficiency, calculated by the inverse of SePCc; ShPUtE (g/mg): Shoot P utilization efficiency, calculated by the inverse of ShPCc; APUtE (g/mg): All P utilization efficiency, calculated by ADWPP divided by APCPP.

**Table 2 ijms-22-09311-t002:** Key candidate genes identified in the low-P, normal-P, and LPTI datasets, and homologous genes annotations in *Arabidopsis thaliana*.

Condition	Traits	Candidate Genes	Description in Maize	Homologous Genes in *Arabidopsis*	Other Names in *Arabidopsis*	Descriptions in *Arabidopsis* in TAIR Website
low-P	ShPUtE	GRMZM2G326707	Phosphate transporter protein1	AT2G38940	*PHT1;4*	The expression is upregulated in the shoot of cax1/cax3 mutant and is responsive to phosphate (Pi) and not phosphite (Phi) in roots and shoots.
	ShPUtE	GRMZM5G848945	Protein AUXIN SIGNALING F-BOX 3	AT3G26810	*AFB2*	The dominant auxin receptor in roots.
				AT1G12820	*AFB3*	Auxin receptor involved in primary and lateral root growth inhibition in response to nitrate. The target of miR393. Induced by nitrate in primary roots.
	SePCPP	GRMZM2G030762	Transcription factor bHLH55	AT3G07340	*CRY2-INTERACTING BHLH 3*	A bHLH DNA-binding superfamily protein.
normal-P	ADWPP	GRMZM2G109967	CDP-diacylglycerol--glycerol-3-phosphate 3-phosphatidyltransferase	AT2G39290	*PHOSPHATIDYLGLYCEROLPHOSPHATE SYNTHASE 1*	Encodes a phosphatidylglycerol phosphate synthase 2C which is dual-targeted into chloroplasts and mitochondria. Mutant plants have mutant chloroplasts but normal mitochondria.
	SePCPP	GRMZM2G418916	Phosphoinositide phosphatase SAC6	AT3G51460	*ROOT HAIR DEFECTIVE4*	A phosphatidylinositol- 4-phosphate phosphatase required for root hair development.
LPTI	ShPCc, ShPUtE	GRMZM2G076630	Probable transcriptional regulator SLK2	AT5G62090	*SLK2*	Encodes a protein that functions with LUH to promote Al binding to the root cell wall.
	YPP	GRMZM2G104125	Calcium- dependent protein kinase 2	AT5G19450	*CDPK19*	Calcium-dependent protein kinase (CDPK19) mRNA, complete.
				AT5G12480	*CPK7*	Calmodulin-domain protein kinase CDPK isoform 7.

## Data Availability

All relevant data are within the paper and its Appendix A.

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
