# Peer review of "Genetic Dissection of Phosphorus Use Efficiency in a Maize Association Population under Two P Levels in the Field"

_ijms, 2021, doi:10.3390/ijms22179311_

Round 1

Reviewer 1 Report

In the manuscript by Li et al, the authors conducted a GWAS analysis for Phosphorus use efficiency in a maize population. Given the importance of maize as a crop worldwide, and the relevance of breeding for high PUE, the paper addresses questions of high relevance worldwide. There are some interesting results therein. However, I believe that there are some points that the authors still need to address. I summarise below major and minor comments.

Major comments:

I am unsure of how useful the gene ontology analysis is in this case. This works well in analysing high number of differentially regulated genes identified using RNASeq for example. However, in this case the genes were identified using a GWAS, which is used to explain the genotypic variance in a set of diverse genotypes. It is possible that some of these are marker traits association are not identified from the same maize genotypes. So, it is possible that some of the functional genes have been identified and linked to different phenotypic response. It doesn’t quite make sense to me to pull all them in a GO enrichment analysis. Besides, it looks like the GWAS threshold value was lower to 4 for this analysis (L245-246), can the author provide some justification?

Phylogenetic analysis of PHT1- Was this analysis done on the complete family of PHT1 in maize? Can the authors discuss their phylogenetic analysis of the PHT1 family? Are there missing members? specific sub-families? Also, it would be interesting to know whether the specific SNPs that allow them to identify the functional gene can be seen in other PHT1?

Though the introduction focuses on PUE, throughout the paper, the emphasis appears to be on PUtE. It is not so clear whether any significant marker traits association had been identified for PUE.

GWAS type of analysis are useful in identifying new components and it is less clear whether in this manuscript it has been the case. It seems that the emphasis has been on SNPs present in genes that have been characterised already and known to be important already.

The manuscript would benefit from a careful editing of the text, to make it easier for the reader. Also, I would rather have the traits name written rather than using abbreviations, too many abbreviations make it difficult to read, especially when these are not so common.

Minor comments:

L25: Add the genus and species name for maize.

L27-29: ‘a significant increase in grain yield per plant and biomass, an increase in PUE’ under add under which treatment.

L30-31: to which analysis does this sentence refer to? It is not so clear.

L46-47: ‘Excessive application and loss’ this sentence could be clarified.

L50: define PUE as ‘the yield produced per unit of P available’.

L103: Define ‘root fork’- is it a measure of root branching?

L114-116: I would suggest rewriting the last sentence and focusing on what the data show. (e.g. not use ‘will provide’ as the results already present interesting information)

Table 1: in the legend, please explain how P utilisation efficiency was calculated.

L120: given that the material and method section is located at the end of the manuscript it would be worth adding some information here about the field trial, just as summary sentence also stating the P levels.

L142-143: Define LPTI index. How was it calculated? It would be good to add this information when LPTI is first cited.

Figure 2: Are all correlations shown statistically significant? If so it could be stated in the legend.

L190-193: SS population appears to show a significantly different values for ShPUtE, APUtE only not SePUtE and also only in comparisons to the NSS population.

L465: Can the authors comment on how population structure was taken into account when running the GWAS?

L419: Can the authors provide additional detailed information about the growth conditions?

Levels of N and K. When was the maize planted? What year? Any additional treatment?

Reviewer 2 Report

  • Line 142: Provide an additional table to show the correlation coefficient mentioned in the text describing Figure 2. It is not clear what the authors are trying to say from Line 142 to Line 159 which clear results were not provided and figure 2 is totally not reader-friendly. Please redo the figure and re-write the paragraph to clarify the message. Also, the authors did not explain what the definition of LPTI is and how it was calculated.
  • Line 194: It is not properly labelled in Figure S3 which the authors do not clarify which figures mean ADWPP and SDWPP.
  • Line 213: Figure 4 was mentioned before Table 2, so please put Figure 4 before Table 2 based on such order.
  • Line 233: The entire 2.3. result section seems cherry-picking there are 49, 53, and 48 candidate genes identified but the authors picked only few of them and the references cited while describing the potential functions of those genes may not be the most proper way. For example: BHLH gene family contains a large number of genes and bHLH32 contributes to P starvation responses in Arabidopsis does not mean bHLH55 in maize does something related to the responses to P stress, not to mention there is no result showing the homology of these two genes.
  • Line 250: re-write the entire first paragraph of the entire 2.4. result section to put GO terms in Figure 5A together and then talk about GO terms in Figure 5B instead of mentioning them repeatedly. Be succinct.
  • Line 269: Suggestion: combine 2.3. and 2.4. result sections as one because the messages look similar.
  • Line 259: I do not see the relevance of putting result section 2.5. in the manuscript. Or the authors need to explain it further into details.
  • Line 278: The authors put three more candidate genes which are not identified in this study in result section 2.6. which makes this study less important! I do not understand.
  • Line 329: do these 20 lines P-efficient lines also contain high APUtE under normal P condition?
  • Line 342: LD: Linkage disequilibrium. This is the first time the authors mention this term in the manuscript, so the authors should use the full name of it instead of the abbreviation.
  • Line 386-389: Hard to catch the point the authors are trying to say here.
  • Overall, the authors did not point out how their results may contribute in helping molecular breeding or how to improve the overall breeding program in order to achieve better PUE.

Round 2

Reviewer 2 Report

The manuscript looks much better now.

  • Line 238: their homologous genes in Arabidopsis were identified
  • Line 295-295: membrane-bound organelle